# CARTS: COOPERATIVE REINFORCEMENT LEARNING FOR TRAFFIC SIGNAL CONTROL AND CARBON EMISSION REDUCTION

## ABSTRACT

Existing traffic signal control systems often rely on overly simplistic, rule-based approaches. Even reinforcement learning (RL)-based methods tend to be suboptimal and unstable due to the inherently local nature of control agents. To address the potential conflicts among these agents, we propose a cooperative architecture named CARTS (CooperAtive Reinforcement learning for Traffic Signal control). CARTS introduces multiple reward terms, weighted with an age-decaying mechanism, to optimize traffic signal control at a global scale. Our framework features two types of agents: local agents that focus on optimizing traffic flow at individual intersections, and a global agent that coordinates across intersections to enhance overall throughput. Importantly, the system is designed to reduce both vehicle waiting time and carbon emissions. We evaluated CARTS using real-world traffic data obtained from traffic cameras in an Asian country. Despite incorporating a global agent during training, CARTS remains decentralized at inference time, requiring no centralized coordination during deployment. Experimental results show that CARTS consistently outperforms state-of-the-art methods across all evaluated performance metrics. Moreover, CARTS effectively links carbon emission reduction with global agent coordination, providing an interpretable and practical approach to sustainable traffic signal control.

## 1 INTRODUCTION

Traffic signal control is a complex and critical real-world problem aimed at minimizing overall vehicle travel time by efficiently coordinating traffic flow at intersections. Traditional traffic signal systems still rely heavily on manually crafted rules, which are often inflexible and incapable of adapting to dynamic traffic patterns or scaling to modern, large-scale transportation networks. Recent advancements in reinforcement learning (RL), particularly deep RL Alemzadeh et al. (2020); Zheng et al. (2019c), offer a promising alternative by enabling agents to learn state representations and policy approximations directly from raw input data. In this paper, we investigate the feasibility of applying RL to on-policy traffic signal control with minimal prior assumptions, paving the way toward more adaptive and intelligent traffic management systems.

Recent studies have proposed various RL-based frameworks for traffic signal control Wei et al. (2021), with most being value-based approaches Zheng et al. (2019c); Mannion et al. (2016); Pham et al. (2013); Van der Pol & Oliehoek (2016); Wei et al. (2018); Arel et al. (2010); Calvo & Dusparic (2018) that converge efficiently but are limited to discrete action, state, and time spaces. Although methods such as Lutter et al. (2021) address value iteration in continuous domains, they have yet to be applied to traffic optimization. In traffic control, this discreteness implies that light phase decisions are bound by pre-defined cyclic sequences (e.g., red/green phases). While discretizing time slots simplifies implementation and enables basic optimization, it fails to adapt to real-time traffic dynamics. Moreover, small fluctuations in the value function can lead to significant shifts in policy. To overcome these issues, recent work has increasingly adopted policy-based RL methods Chu et al. (2019); Nishi et al. (2018); Mousavi et al. (2017), which support continuous phase durations. However, these approaches are heavily sample-dependent and prone to convergence to suboptimal solutions.

Figure 1: Overview of CARTS. (a) Architecture used in the training stage. A main novelty of CARTS is the introduction of a global agent highlighted in red. (b) Architecture used in the inference stage. The global agent is not required for performing. Thus, the overall system remains decentralized.

To bridge value-based and policy-based reinforcement learning (RL), the actor-critic framework—especially Deep Deterministic Policy Gradient (DDPG)Lillicrap et al. (2015)—is widely used for stabilizing training by learning both a $Q$-function and a deterministic policyPang & Gao (2019); Wu (2020). However, DDPG often converges to local optima and lacks global coordination. Most RL-based traffic signal control methods Pang & Gao (2019); Wu (2020); Chu et al. (2019); Nishi et al. (2018); Mousavi et al. (2017) rely on decentralized agents using only local rewards, often leading to conflicts across intersections. Additionally, few frameworks Aslani et al. (2017; 2018) address phase duration explicitly, typically using fixed time slots. CoSLight Ruan et al. (2024) improves coordination via dynamic collaborator selection, showing strong results. Still, predefined-duration approaches remain less adaptive than on-demand solutions for congestion relief.

We propose CARTS (CooperAtive Reinforcement Learning for Traffic Signal control), a novel framework designed to optimize traffic signal control while simultaneously reducing carbon emissions (see Fig. 1). The core innovation of CARTS is the introduction of a global agent that collaborates with local agents by balancing their individual objectives to improve overall traffic throughput. Each local agent learns a policy based on intersection-level rewards, such as clearance efficiency. In parallel, the global agent optimizes a higher-level objective-"total traffic waiting time" by aggregating information across intersections using an actor-critic framework. CARTS selects optimal traffic signal phases and determines their dynamic durations to maximize flow efficiency. Importantly, the global agent is used only during training, making the system fully decentralized at inference. To ensure scalability, it receives information solely from nearby intersections. Unlike prior policy-based approaches that rely on fixed-duration action pools, CARTS jointly learns both the optimal phase and its variable length. Furthermore, the framework explicitly targets carbon emission reduction, a feature not addressed in existing work. Extensive experiments on real-world traffic datasets and the benchmark from Ault & Sharon (2021) show that CARTS significantly outperforms state-of-the-art methods in both traffic efficiency and environmental impact.

- We introduce CARTS, a cooperative RL framework that effectively reduces congestion, minimizes travel time, and increases network-wide traffic throughput.

- We introduce a global agent that resolves conflicts during training and steers local agents toward coherent policies, while keeping inference fully decentralized.

- CARTS supports dynamic phase durations, moving beyond fixed action durations used in SoTA methods.

- Our framework is the first to link carbon emissions with global coordination, demonstrating significant reductions in both waiting time and $CO_2$ emissions.

- Comprehensive experiments on real-world data and public benchmarks confirm that CARTS achieves state-of-the-art performance in traffic signal control.

## 2 RELATED WORK

**Traditional traffic control** methods fall into three categories: (1) fixed-time control Roess et al. (2004), (2) actuated control Fellendorf (1994); Mirchandani & Head (2001), and (3) adaptive control Zheng et al. (2019a;c). These methods rely heavily on human expertise to design signal cycles and strategies, making parameter tuning labor-intensive and inflexible across varying conditions

such as peak, off-peak, and normal hours. Fixed-time control is widely adopted for its simplicity, using preset schedules regardless of traffic flow. Actuated control responds to real-time traffic conditions based on predefined thresholds—for example, triggering a green phase when queue lengths exceed a certain limit. Adaptive control methods, such as SCATS Lowrie (1990), dynamically select signal phases based on live traffic data and offer improved optimization compared to static approaches.

**RL traffic control:** Recent advances in reinforcement learning (RL) offer promising improvements for automated traffic signal control. RL methods generally fall into three categories: value-based, policy-based, and actor-critic (AC) approaches Aslani et al. (2017). Value-based methods, like $Q$-learning Watkins & Dayan (1992), estimate expected returns for state-action pairs and derive policies accordingly. Early traffic signal applications include Abdoos et al. (2011), but maintaining full $Q$-tables becomes infeasible for large state-action spaces. Deep Q-Networks (DQNs)Guo et al. (2014) mitigate this but suffer from overestimation, which Double DQNVan Hasselt et al. (2016) resolves using separate networks for selection and evaluation. While fast to converge, value-based methods are limited to discrete actions. In contrast, policy-based methods use policy gradients to support continuous control and better adaptability but often require longer training due to round-based updates.

RL methods can also be categorized by action type: (i) setting green-light duration, (ii) deciding whether to switch phases, and (iii) selecting the next phase. Value-based and AC methods are better suited for (ii) and (iii), which involve discrete choices. However, they are less ideal for (i), where continuous phase durations are required. DDPG Lillicrap et al. (2015) is well-suited for this setting and has inspired various DDPG-based traffic control frameworks Pang & Gao (2019); Wu (2020). Most of these, however, are limited to single intersections. In practice, traffic signal control requires coordination across many intersections. To handle this, multi-agent DDPG algorithms Gupta et al. (2017); Lowe et al. (2017) have been proposed to incorporate inter-agent communication. Notably, CityFlow Zhang et al. (2019) simulates large-scale urban traffic and applies MARL for control. However, existing multi-agent frameworks still lack the ability to output dynamic phase durations, limiting their ability to communicate precise phase timing to drivers in real-world deployments.

Recent frameworks have begun integrating domain knowledge into large language models (LLMs) to enhance global traffic signal control (TSC) capabilities. For instance, LA-LightWang et al. (2024) incorporates an LLM agent to leverage commonsense reasoning for tool selection, enabling more informed decision-making in complex scenarios such as emergencies or equipment failures. Similarly, PromptGATDa et al. (2024) uses prompt-based grounded action transformation to incorporate LLM knowledge and address the sim-to-real gap in traffic control. However, these approaches typically relegate LLMs to a supplementary role, limiting their potential for autonomous, globally-aware decision-making. Moreover, for real-time applications, latency introduced by LLM inference poses a significant challenge compared to traditional RL agents.

## 3 BACKGROUND AND NOTATIONS

The basic elements of an RL problem for traffic signal control can be formulated as a Markov Decision Process (MDP) mathematical framework of $< S, A, T, R, \gamma >$, with the following definitions:

- $S$ denotes the set of states, which is the set of all lanes containing all possible vehicles. $s_t \in S$ is a state at time step $t$ for an agent.
- $A$ denotes the set of possible actions, which is the duration of the green light. In our scenarios, both the durations of a traffic cycle and a yellow light are fixed. Then, once the state of the green light is chosen, the duration of the red light can be determined. At time step $t$, the agent can take an action $a_t$ from $A$.
- $T$ denotes the transition function, which stores the probability of an agent transiting from state $s_t$ to $s_{t+1}$ if the action $a_t$ is taken; that is, $T(s_{t+1}|s_t, a_t) : S \times A \rightarrow S$.
- $R$ denotes the reward, where at time step $t$, the agent obtains a reward $r_t$ specified by a reward function $R(s_t, a_t)$ if the action $a_t$ is taken under state $s_t$.
- $\gamma$ denotes the discount, which controls the importance of the immediate reward versus future rewards, and also ensures the convergence, where $\gamma \in [0, 1)$.

At time-step $t$, the agent determines its next action $a_t$ based on the current state $s_t$. After executing $a_t$, it will transit to next state $s_{t+1}$ and receive a reward $r_t(s, a)$; that is, $r_t(s, a) = \mathbb{E}[R_t | s_t = s, a_t = a]$, where $R_t$ is named as the one-step reward. The way that the RL agent chooses an action is named policy and denoted by $\pi$. The policy is a function $\pi(s)$ that chooses an action from the current state $s$; that is, $\pi(s) : S \rightarrow A$. Our goal is to find such a policy to maximize the future reward $G_t$:

$$G_t = \Sigma_{k=0}^{\infty} \gamma^k R_{t+k}. \tag{1}$$

**Deep Q-Network (DQN):** In Hester et al. (2018); Van Hasselt et al. (2016), a deep neural network is used to approximate the $Q$ function, which enables the RL algorithm to learn $Q$ well in high-dimensional spaces. Let $Q_{tar}$ be the true target value expressed as $Q_{tar} = r + \gamma \max_{a'} Q(s', a'; \theta)$.

Also, let $Q(s, a; \theta)$ be the estimated value, where $\theta$ is the set of its parameters. We define the loss function for training the DQN as:

$$L(\theta) = \mathbb{E}_{s,a,r,s'}[(Q_{tar} - Q(s, a; \theta))^2]. \tag{2}$$

In Mnih et al. (2015), $Q_{tar}$ is often overestimated during training and results in the problem of unstable convergence of the $Q$ function. In Van Hasselt et al. (2016), a Double DQN (DDQN) was proposed to address this unstable problem by separating the DDQN into two value functions, so there are two sets of weights $\theta$ and $\phi$ to parameterize the original value function and the second target network, respectively. The second DQN $Q_{tar}$ with parameters $\phi$ is a lagged copy of the first DQN $Q(s, q; \theta)$ that can fairly evaluate the $Q$ value as follows:

$$Q_{tar} = r + \gamma Q(s', \max_{a'} Q(s', a'; \theta); \phi) \tag{3}$$

**Deep Deterministic Policy Gradient (DDPG)**: DDPG is a model-free, off-policy reinforcement learning framework that leverages deep neural networks for function approximation. Unlike DQN, which is limited to discrete and low-dimensional action spaces, DDPG is designed to handle continuous action spaces. As an actor-critic method, it consists of both a policy network (actor) and a value function network (critic). The critic network in DDPG is similar to that in traditional actor-critic architectures. Inspired by Double DQN Van Hasselt et al. (2016), DDPG improves robustness by maintaining two separate networks (target and online) to estimate value functions and reduce overestimation bias. In our work, we adopt DDPG to jointly learn both the optimal $Q$-function and its corresponding policy in continuous control settings.

## 4 THE PROPOSED METHOD

he original DDPG leverages off-policy data and the Bellman equation Barron & Ishii (1989) to learn the $Q$-function, from which it derives the policy. It alternates between approximating the optimal value function $Q^($s, a)$ and determining the corresponding optimal action $a^($s)$, enabling learning in continuous action spaces. In DDPG, the output is a continuous-valued action, which in our case represents the duration (in seconds) of the green light. Although DDPG is inherently off-policy, it can incorporate past experiences and current environment parameters by interacting with traffic simulators—such as TSISOwen et al. (2000) or SUMOKrajzewicz et al. (2002)—to generate diverse training data, including on-policy samples. For consistency and fair comparison with state-of-the-art methods, which predominantly use SUMO, we also conduct all our experiments and ablation studies using SUMO exclusively.

The main novelty of this paper lies in introducing a cooperative learning mechanism with a global agent to guide local agents and prevent them from blindly exploring the environment, thereby significantly improving both learning effectiveness and overall traffic throughput (see Fig. 5). In standard DDPG, exploration is encouraged by adding random noise to the action outputs. While this promotes environmental exploration, it can lead to inefficient and uncoordinated behavior, especially in multi-agent traffic signal control, where each intersection is controlled by a separate local agent. Existing methods typically apply the same noise-based strategy to every agent, which increases the likelihood of conflicting decisions between neighboring intersections during training. Blind exploration not only hampers convergence but also reduces throughput, as conflicting local actions can degrade overall system performance. To address this, we propose integrating a global agent into

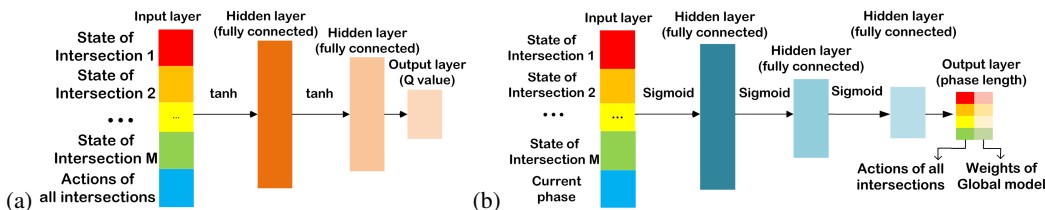

Figure 2: **Architectures for the global agent.** (a) The **global critic network** calculates the predicted total waiting time throughout the site. (b) The **global actor network** calculates a vector containing the actions and weight vectors at all intersections.

the DDPG framework to coordinate local agents during training. This cooperative mechanism helps align local decisions with global goals, resulting in more efficient and conflict-free policy learning.

### 4.1 COOPERATIVE DPGG NETWORK ARCHITECTURE

Most policy-based RL methods Chu et al. (2019); Nishi et al. (2018); Mousavi et al. (2017) rely on local agents to control intersections, which can cause conflicting objectives and slow convergence. To address this, we propose CARTS, where each intersection is managed by a local agent but guided by a global agent during training (Fig. 1). The global agent coordinates local policies to optimize system-wide rewards, then is removed at inference. Trained agents operate autonomously, using observations from all intersections to make real-time signal decisions.

Although the DDPG method is off-policy, we use TSIS Owen et al. (2000) and SUMO Krajzewicz et al. (2002) to collect on-policy data for RL training. With the on-policy data, the parameters of local and global agents are updated by the **Local Agent Updating (LAU)** algorithm and **Global Agent Updating (GAU)** algorithm, respectively. Let $W_G^m$ represent the global agent's importance to the $m$-th intersection. Then, the importance $W_L^m$ of the $m$-th local agent will be 1-$W_G^m$. For the $m$-th intersection, its next action will be predicted by **Generating On-policy Data (GOD)** and LAU algorithms, respectively, via an epsilon greedy exploration scheme. The output seconds of the global agent and the local agent are compared based on $W_G^m$ and $W_L^m$. Then, the one with higher importance will be chosen for the output seconds. Details of our algorithm are provided in Algorithm 1 (see the appendix).

### 4.2 GENERATING ON-POLICY DATA

During the RL-based training process, before starting, we will perform a one-hour simulation to collect data (see detailed algorithm in the supplementary material) and store them in the replay buffer $\mathbf{B}$ based on TSIS or SUMO. Let $\mathbf{B_m}$ be the set o f on-policy data collected for training the $m$-th local agent. Then $\mathbf{B}$ is the union of all $\mathbf{B_m}$, *i.e.*, $\mathbf{B}= (\mathbf{B_1}, ..., \mathbf{B_m}, ..., \mathbf{B_M})$. In the process of interacting with the environment, we will add the $\epsilon$-greedy and weight-decayed method to the selection of actions. In particular, the $\epsilon$-greedy method will gradually reduce $\epsilon$ from 0.9 to 0.1 and a time decay mechanism is adopted to decay $W_G^m$ by the ratio $(0.95)^t$ in the $t$-th iteration.

### 4.3 THE LOCAL AGENTS

In our simulation, each intersection is assigned a fixed cycle duration for traffic signal changes. Additionally, the yellow light duration is fixed at $Y$ seconds. Thus, the model only needs to determine the green light phase duration, while the red light duration can be directly derived from the cycle length. At each intersection, a DDPG-based architecture is employed to represent the local agent responsible for traffic control. The following definitions describe this local agent.

1. The duration of the traffic phase ranges from $D_{min}$ to $D_{max}$ seconds.

2. Stopped vehicles are defined as vehicles whose speeds are less than 3 $km/hr$.

3. The state at an intersection is defined by a vector in which each entry records the number of vehicles stopped in each lane at this intersection at the end of the green light and the current traffic signal phase.

The reward for evaluating the quality of a state at an intersection is based on the degree of clearance, i.e., the number of vehicles remaining when the green-light phase ends. Rewards are assigned in two scenarios: (1) The green light ends but vehicles remain. (2) The green light is still active, but no vehicles are present. No rewards or penalties are given in other situations. Let $N_{m,t}$ denote the number of vehicles at intersection $m$ at time $t$, and let $N_{max}$ be the maximum traffic flow. We use the clearance degree as a reward for qualifying the $m$th local agent. When the green light ends and there is no vehicle, a pre-defined max reward $R_{max}$ is assigned to the $m$-th local agent. If there are still some vehicles, a penalty proportional to $N_{m,t}$ is assigned to this local agent. More precisely, for Case 1, the reward $r_{m,t}^{local}$ for the intersection $m$ is defined as:

**Case 1**: If the green light ends but some vehicles are still,

$$r_{m,t}^{local} = \begin{cases} R_{max}, if \ \frac{N_{m,t}}{N_{max}} \leq \frac{1}{N_{max}}; \\ -\frac{R_{max}N_{m,t}}{N_{max}}, \text{else.} \end{cases} \tag{4}$$

For Case 2, if there is no traffic but a long period still remaining for the green light, various vehicles moving on another road should stop and wait until this green light turns off. To avoid this case, a penalty should be given to this local agent. Let $g_{m,t}$ denote the remaining green light time (counted by seconds) when there is no traffic flow in the $m$-th intersection at time step $t$, and $G_{max}$ the longest green light duration. Then, the reward function for Case 2 is defined as:

**Case 2**: If there is no traffic but the green light is still on,

$$r_{m,t}^{local} = \begin{cases} R_{max}, if \ \frac{g_{m,t}}{G_{max}} \leq \frac{1}{G_{max}}; \\ -\frac{R_{max}g_{m,t}}{G_{max}}, \text{else.} \end{cases} \tag{5}$$

Detailed architectures for local agents are shown in the supplementary material. Its inputs are the number of vehicles stopped at the end of the green light in each lane, the remaining seconds of the green light, and the current phases of the traffic signals at all intersections. Thus, the input dimension for each local critic network is $\left(2M + \sum_{m=1}^{M} N_m\right)$, where $M$ denotes the number of intersections and $N_m$ is the number of lanes at the $m$-th intersection. Then, a hyperbolic tangent function is used as an activation function to normalize all input and output values. There are two fully connected hidden layers to model the $Q$-value. The output is the expected value of a future return of taking this action in the state. The inputs used to model the local actor network include the number of stopped vehicles at the end of the green light at each lane, and the current traffic signal phases of all intersections. Thus, the dimension for each local actor network is $(M + \sum_{m=1}^{M} N_m)$. Let $\theta_m^Q$ and $\theta_m^\mu$ denote the sets of parameters of the $m$th local critic and actor networks, respectively. To train $\theta_m^Q$ and $\theta_m^\mu$, we sample a random mini-batch of $N_b$ transitions $(\boldsymbol{S}_i, \boldsymbol{A}_i, \boldsymbol{R}_i, \boldsymbol{S}_{i+1})$ from $\mathbf{B}$, where

1. Each state $\boldsymbol{S}_i$ is an $M \times 1$ vector containing the local states of all intersections;
2. Each action $\boldsymbol{A}_i$ is an $M \times 1$ vector containing the seconds of current phase of all intersections;
3. Each reward $\boldsymbol{R}_i$ is an $M \times 1$ vector containing the rewards obtained from each intersection after taking $\boldsymbol{A}_i$ at the state $\boldsymbol{S}_i$. The $m$-th entry of $\boldsymbol{R}_i$ is the reward of the $m$th intersection after taking $\boldsymbol{A}_i$.

Let $y_i^m$ denote the reward after taking action $\boldsymbol{A}_i$ from the $m$-th target critic network. Based on $y_i^m$, the loss functions for updating $\theta_m^Q$ and $\theta_m^\mu$ are defined, respectively, as follows:

$$L_{critic}^m = \frac{1}{N_b} \sum_{i=1}^{N_b} (y_i^m - Q(\boldsymbol{S}_i, \boldsymbol{A}_i | \theta_m^Q))^2 \ \text{ and}$$

$$L_{actor}^m = -\frac{1}{N_b} \sum_{i=1}^{N_b} Q(\boldsymbol{S}_i, \mu(\boldsymbol{S}_i | \theta_m^\mu) | \theta_m^Q). \tag{6}$$

With $\theta_m^Q$ and $\theta_m^\mu$, the parameters $\theta_m^{Q'}$ and $\theta_m^{\mu'}$ for the target network are constantly updated as follows:

$$\theta_m^{Q'} \leftarrow (1 - \tau)\theta_m^Q + \tau\theta_m^{Q'} \ \text{ and } \ \theta_m^{\mu'} \leftarrow (1 - \tau)\theta_m^\mu + \tau\theta_m^{\mu'}. \tag{7}$$

The parameter $\tau$ is set to 0.8 to update the target network. Refer to the supplementary material for the detailed algorithm to update the parameters of local agents.

| Methods | I-1 | I-2 | I-3 | I-4 | I-5 | Average |
|---|---|---|---|---|---|---|
| Fixed | 1,530 | 1,560 | 1,996 | 2,288 | 2,291 | 1,933 |
| MA-DDPG | 1,782 | 1,819 | 2,098 | 1,896 | 2,400 | 1,999 |
| PPO | 979 | 957 | 1,206 | 1,517 | 1,619 | 1,255.6 |
| TD3 | 1,370 | 1,394 | 1,787 | 2,070 | 2,147 | 1,753.6 |
| **CARTS (Ours)** | **2,225** | **2,310** | **2,784** | **3,052** | **2,868** | **2,647.8** |

Table 1: Comparisons of throughput against other SoTA methods. Best scores are marked in bold.

| Methods | Delay | Speed | Time loss | Travel time | Wait time |
|---|---|---|---|---|---|
| IDQN | 2,745.96 | 11.01 | 258.99 | 227.67 | 217.78 |
| IPPO | 2,463.69 | 9.62 | 1,576.62 | 236.49 | 1,538.05 |
| FMA2C | 2,734.2 | 11.19 | 151.12 | 226.56 | 69.95 |
| MPLight | 2,712.2 | 11.19 | 158.55 | 226.53 | 73.71 |
| MPLight* | 2,709.93 | 11.19 | 186.94 | 226.51 | 90.81 |
| CART | **522** | **14.56** | **138.4** | **156.56** | **38.4** |

\* MPLight full+IDQN

Table 2: Performance comparisons on **two** intersections. Best scores are marked in bold.

## 4.4 THE GLOBAL AGENT

To resolve potential conflicts among local agents, we introduce a global agent that explores the environment with a broader perspective. The global agent optimizes the total waiting time across all intersections. Fig. 5 illustrates the global critic and actor networks; detailed architectures are provided in the supplementary material. To address the high memory demand of a large global agent, we propose a localized global agent that considers only nearby intersections, since information from distant intersections is less relevant. Specifically, we implement global agents covering $3 \times 3$ and $5 \times 5$ grids, as shown in Fig. 3. For example, the $5 \times 5$ grid centers on a blue dot representing the target intersection, and this window can be shifted to cover other intersections as needed. For the $m$-th intersection, we use $V_m$ to denote the number of total vehicles, and $T_{m,n}^{w,i}$ the waiting time of the $n$-th vehicle at time step $i$. Then, the total waiting time on the entire site is used to define the global reward as follows: $r_i^G = -\frac{1}{M} \sum_{m=1}^{M} \sum_{n=1}^{V_m} T_{m,n}^{w,i}$. Let $\theta_G^Q$ and $\theta_G^\mu$ denote the parameters of the global critic and actor networks, respectively. To train $\theta_G^Q$ and $\theta_G^\mu$, we sample a random minibatch of $N_b$ transitions $(\boldsymbol{S}_i, \boldsymbol{A}_i, \boldsymbol{R}_i, \boldsymbol{S}_{i+1})$ from B. Let $y_i^G$ denote the reward after performing $A_i$ obtained from the global target critic network. Then, the loss function for updating $\theta_G^Q$ is defined as follows:

$$L_{critic}^G = \frac{1}{N_b} \sum_{i=1}^{N_b} (y_i^G - Q_G(\boldsymbol{S}_i, \boldsymbol{A}_i | \theta_G^Q))^2. \tag{8}$$

Note that the output of this global critic network is a scalar value, *i.e.*, the predicted total waiting time throughout the site. To train $\theta^{\boldsymbol{\mu}_G}$, we use the following loss function:

$$L_{actor}^G = -\frac{1}{N_b} \sum_{i=1}^{N_b} Q_G(\boldsymbol{S}_i, \boldsymbol{\mu_G}(\boldsymbol{S}_i | \theta^{\boldsymbol{\mu}_G}) | \theta_G^Q). \tag{9}$$

The output of the global actor network is an $M \times 1$ vector, representing the suggested actions for all intersections. The weight $W_G^m$ denotes the importance of the $m$-th intersection within the global agent's scope. Both local and global agents are modeled using DDPG networks. For detailed algorithmic steps on updating the global agent, please refer to the supplementary material. We utilize the TSIS and SUMO simulation platforms to model traffic flow, generating various vehicles that traverse intersections of different scales.

## 4.5 CARBON EMISSION REDUCTION

Another key contribution of this work is the reduction of carbon emissions in traffic signal control. For the first time, we integrate the HBEFA emission model, built into the SUMO simulation platform, to record and output real-time data on vehicle fuel consumption and carbon emissions. The calculation formulas for CO emissions are provided below, with detailed parameter explanations in

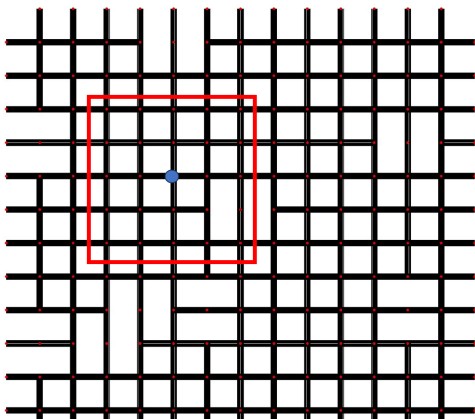

Figure 3: Our lightweight global agent at the size of $5 \times 5$.

| Methods | Delay | Speed | Time loss | Travel time | Wait time |
|---------|-------|-------|-----------|-------------|-----------|
| IDQN | 1,527.49 | 10.72 | 715.76 | 264.53 | 201.27 |
| IPPO | 1,789.1 | 7.27 | 1,268.89 | 346.06 | 658.61 |
| FMA2C | 1,434.68 | 10.72 | 275.69 | 167.96 | 254.54 |
| MPLight | 1,489.91 | 10.61 | 322.02 | 243.11 | 260.19 |
| MPLight* | 1,778.32 | 9.32 | 931.36 | 280.76 | 281.14 |
| CARTS | **466.05** | **10.75** | **115.74** | **113.25** | **134.89** |

\* MPLight full+IDQN

Table 3: Performance comparison on **five** intersections. Best scores are marked in bold.

the supplementary material. The formula for $CO_2$ emissions follows a similar structure, with CO replaced by $CO_2$.

$$CO_{move} = \frac{CO_{engine} \times V_{engine} \times FC \times M_{fuel}}{M_{air} \times 1000}, \tag{10}$$

$$CO_{stop} = \frac{CO_{engine} \times V_{engine} \times r_{stop} \times t_{stop}}{3600 \times M_{air}}, \tag{11}$$

$$CO = CO_{move} + CO_{stop}, \tag{12}$$

where $CO$ and $CO_2$ denote the vehicle emissions of carbon monoxide and carbon dioxide, measured in grams under driving conditions. $CO_{engine}$ and $CO_{2_{engine}}$ represent the respective emissions from the engine, measured in grams per kilowatt-hour (g/kWh). $V_{engine}$ is the engine displacement in liters, while FC denotes fuel consumption in liters per 100 kilometers (L/100km). $M_{fuel}$ and $M_{air}$ are the molecular weights of the fuel and air, measured in grams per mole, with $M_{air}$=28.97 g/mol. The value of $M_{fuel}$ depends on the fuel type. According to the HBEFA formula, the primary influencing factors are the distance traveled (v) and the waiting time ($t_{stop}$). Since $t_{stop}$ during our experiments, the waiting time is the main variable affecting emissions. Thus, reducing $t_{stop}$ directly decreases carbon emissions.

## 5 EXPERIMENTAL RESULTS

Our traffic dataset consists of visual monitoring sequences from five consecutive intersections during the morning rush hour in a midsize Asian city. Experiments simulating real traffic flows were conducted using two simulators: TSISOwen et al. (2000) and SUMO. A fixed-time traffic signal control with a total one-hour waiting time served as the baseline for comparison. We performed ablation studies to evaluate the impact of the global agent within the CARTS framework. Additionally, we benchmarked our method against an open datasetAult & Sharon (2021) for fair comparison with other state-of-the-art approaches.

Table 1 reports throughput across five intersections for fixed-time control, MA-DDPG, PPO, TD3, and CARTS. The global agent notably boosts throughput beyond both baselines and MA-DDPG.

| Methods | Travel time | Avg. wait time | Speed | Fuel | $CO$ | $CO_2$ |
|---|---|---|---|---|---|---|
| No global agent | 1,857.74 | 269.57 | 7.97 | 1 | 109.12 | 2,208.37 |
| W/ global agent | 1,680.89 | 217.51 | 9.2 | 0.93 | 107.86 | 2,160.67 |

Table 4: Ablation study of our method on a real-world site with 16 intersections. Unit for Fuel, $CO$, and $CO_2$ is $mg/s$.

| Ablation Study | 49 intersections | | | | 169 intersections | | | |
|---|---|---|---|---|---|---|---|---|
| Methods | Avg. wait time | Fuel | $CO$ | $CO_2$ | Avg. wait time | Fuel | $CO$ | $CO_2$ |
| No global agent | 301.22 | 1.03 | 114.57 | 2,334.7 | 659.2 | 1.08 | 114.63 | 2,511.99 |
| Using global agent of $3 \times 3$ | 224.43 | 0.95 | 106.24 | 2,206.69 | 528.77 | 1.03 | 107.3 | 2,400.36 |
| Using global agent of $5 \times 5$ | 217.51 | 0.94 | 105.9 | 2,190.12 | 494.53 | 1 | 103.35 | 2,333.42 |

Table 5: Ablation study of our method on the simulation with 49 and 169 intersections to simulate traffic conditions in a large city. Unit for Fuel, $CO$, and $CO_2$ is $mg/s$.

For larger networks (¿10 intersections), processing all nodes is impractical, so the global agent is restricted to eight nearby intersections during training. For fair benchmarking, we use the SUMO-based RL testbed Ault & Sharon (2021), which supports single- and multi-agent traffic control with OpenAI Gym integration and open-source data/code. We evaluate five SoTA baselines: IDQN Ault & Sharon (2021), IPPO Ault & Sharon (2020), FMA2C Chu et al. (2016), MPLight Zheng et al. (2019b), and MPLight-full Ault & Sharon (2021); Zheng et al. (2019b). IDQN and IPPO are decentralized; FMA2C scales to large multi-agent settings; MPLight models phase competition; and MPLight-full augments MPLight with pressure-state sensing. Each method applies its own state and reward design.

Table 2 compares CARTS with existing methods under a two-intersection setting. IDQN Ault & Sharon (2021) and IPPO, both independent-agent approaches, underperform multi-agent methods like FMA2C and MPLight. While MPLight Zheng et al. (2019b) improves coordination via pressure dynamics, it still lags behind FMA2C and CARTS. Our framework uses a global agent during training to guide local agents, then removes it at inference to remain decentralized. This design enhances coordination and policy learning, enabling CARTS to consistently surpass all SoTA baselines across every metric.

To test scalability, we extended evaluation to five intersections (Tab. 3). IPPO remains unstable, especially in Time Loss, while MPLight-full offers modest gains and IDQN Ault & Sharon (2021) improves in Speed and Waiting Time. FMA2C performs strongly across several metrics, but CARTS consistently outperforms all baselines, confirming its robustness, coordination efficiency, and scalability for larger traffic networks.

**Ablation Study.** Table 4 presents ablation study results for a 16-intersection scenario, evaluating Travel Time, Average Wait Time, Speed, $CO$, $CO_2$ and Fuel Consumption. We use a real-world map composed of multiple major junctions arranged in a $4 \times 4$ checkerboard pattern. The results show that our method performs significantly better with the inclusion of the global agent. Additionally, following the HBEFA model integrated in SUMO, emissions of $CO_2$ and fuel consumption are also reduced. To test the scalability of our architecture to more than one hundreds of intersections, we further evaluated larger networks of size $7 \times 7$ and $13 \times 13$, comparing different global agent sizes, where there are 169 intersections in the $13 \times 13$ case. In addition to $3 \times 3$ agent, we explored a larger $5 \times 5$ configuration. As shown in Table 5, increasing the size of the global agent leads to consistent improvements in overall system performance.

## 6 CONCLUSION

We presented CARTS, a novel cooperative RL architecture that well handles the cooperation problems for traffic signal control among local agents by adding a global agent. The global agent has access to all intersection information to guide local agents to improve training. CARTS remains decentralized despite the inclusion of a global agent, so the global agent is not required for performing inference. Using SUMO traffic simulation, we have shown how CARTS significantly improves system throughput and reduce carbon emissions.

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

# A APPENDIX

In this section, we will prove that value function in our method will actually converge.

**Definition A.1.** *A metric space $< M, d >$ is complete (or Cauchy) if and only if all Cauchy sequences in $M$ will converge to $M$. In other words, in a complete metric space, for any point sequence $a_1, a_2, \cdots \in M$, if the sequence is Cauchy, then the sequence converges to $M$:*

$$\lim_{n \to \infty} a_n \in M.$$

**Definition A.2.** *Let (X,d) be a complete metric space. Then, a map $T : X \to X$ is called a contraction mapping on X if there exists $q \in [0, 1)$ such that $d(T(x), T(y)) < qd(x, y), \forall x, y \in X$.*

**Theorem 1** (Banach fixed-point theorem)**.** *Let (X,d) be a non-empty complete metric space with a contraction mapping $T : X \to X$. Then T admits a unique fixed-point $x^*$ in X. i.e. $T(x^*) = x^*$.*

**Theorem 2** (Gershgorin circle theorem)**.** *Let A be a complex $n \times n$ matrix, with entries $a_{ij}$. For $i \in 1, 2, ..., n$, let $R_i$ be the sum of the absolute of values of the non-diagonal entries in the $i^{th}$ row:*

$$R_i = \sum_{j=0, j \neq i}^{n} |a_{ij}|.$$

*Let $D(a_{ii}, R_i) \subseteq \mathbb{C}$ be a closed disc centered at $a_{ii}$ with radius $R_i$, and every eigenvalue of A lies within at least one of the Gershgorin discs $D(a_{ii}, R_i)$.*

**Lemma 3.** *We claim that the value function of RL can actually converge, and we also apply it to traffic control.*

*Proof.* The value function is to calculate the value of each state, which is defined as follows:

$$\begin{aligned}
V^\pi(s) &= \sum_a \pi(a|s) \sum_{s', r} p(s', r|s, a)[r + \gamma V^\pi(s')] \\
&= \sum_a \pi(a|s) \sum_{s', r} p(s', r|s, a)r \\
&+ \sum_a \pi(a|s) \sum_{s', r} p(s', r|s, a)[\gamma V^\pi(s')].
\end{aligned} \tag{13}$$

Since the immediate reward is determined, it can be regarded as a constant term relative to the second term. Assuming that the state is finite, we express the state value function in matrix form below. Set the state set $S = \{S_0, S_1, \cdots, S_n\}$, $V^\pi = \{V^\pi(s_0), V^\pi(s_1), \cdots, V^\pi(s_n)\}^T$, and the transition matrix is

$$P^\pi = \begin{pmatrix} 0 & P^\pi_{0,1} & \cdots & P^\pi_{0,n} \\ P^\pi_{1,0} & 0 & \cdots & P^\pi_{1,n} \\ \cdots & \cdots & \cdots & \cdots \\ P^\pi_{n,0} & P^\pi_{n,1} & \cdots & 0 \end{pmatrix}, \tag{14}$$

where $P^\pi_{i,j} = \sum_a \pi(a|s_i)p(s_j, r|s_i, a)$. The constant term is expressed as $R^\pi = \{R_0, R_1, \cdots, R_n\}^T$. Then we can rewrite the state-value function as:

$$V^\pi = R^\pi + \lambda P^\pi V^\pi. \tag{15}$$

Above we define the state value function vector as $V^\pi = \{V^\pi(s_0), V^\pi(s_1), \cdots, V^\pi(s_n)\}^T$, which belongs to the value function space $V$. We consider $V$ to be an n-dimensional vector full space, and define the metric of this space is the infinite norm. It means:

$$d(u, v) = \| u - v \|_\infty = \max_{s \in S} |u(s) - v(s)|, \forall u, v \in V \tag{16}$$

Since $< V, d >$ is the full space of vectors, $V$ is a complete metric space. Then, the iteration result of the state value function is $u_{new} = T^\pi(u) = R^\pi + \lambda P^\pi u$. We can show that it is a contraction mapping.

$$\begin{aligned}
d(T^\pi(u), T^\pi(v)) &= \| (R^\pi + \lambda P^\pi u) - (R^\pi + \lambda P^\pi v) \|_\infty \\
&= \| \lambda P^\pi(u - v) \|_\infty \\
&\leq \| \lambda P^\pi \| u - v \|_\infty \|_\infty .
\end{aligned} \tag{17}$$

From Theorem 2, we can show that every eigenvalue of $P^\pi$ is in the disc centered at $(0,0)$ with radius 1. That is, the maximum absolute value of eigenvalue will be less than 1.

$$
\begin{aligned}
d(T^\pi(u), T^\pi(v)) \leq &\parallel \lambda P^\pi \parallel \parallel u - v \parallel_\infty \parallel_\infty \\
\leq &\lambda \parallel u - v \parallel_\infty \\
= &\lambda d(u, v).
\end{aligned}
\tag{18}
$$

From the Theorem 1, Eq.(2) converges to only $V^\pi$. $\qquad\square$

## B    ALGORITHM

---

**Algorithm 1** CARTS: CooperAtive Reinforcement Learn- ing for Traffic Signal control

---

Initialize critic network $Q(s,a|\theta^Q)$ and actor network $\mu(s|\theta^\mu)$ with random weights $\theta^Q$ and $\theta^\mu$.

Initialize target network $Q'$ and $\mu'$ with weights $\theta^{Q'} \leftarrow \theta^Q, \theta^{\mu'} \leftarrow \theta^\mu$ and also initialize replay buffer $R$.

**for** *t=1, ... ,T* **do**

    Clean the replay buffer **B**.

    /* $\mathbf{B} = (\mathbf{B_1}, ..., \mathbf{B_m}, ..., \mathbf{B_M})$; */

    /* $B^m$: on-policy data for the $m$-th intersection */

    /* Generate on-policy data */

    $\mathbf{B} = GOD(t)$;

    **for** *episode=1, ..., 400* **do**

        **for** *m=1,..., M, Global* **do**

            **if** $m \neq Global$ **then**

                $LAU(\mathbf{B},m)$;// Update local agents

            **if** *agent=Global* **then**

                $GAU(\mathbf{B})$;// Update the global agent

---

**Algorithm 2** COMA-DDPG traffic signal control RL algorithm.

---

Initialize critic network $Q(s,a|\theta^Q)$ and actor network $\mu(s|\theta^\mu)$ with random weights $\theta^Q$ and $\theta^\mu$.

Initialize target network $Q'$ and $\mu'$ with weights $\theta^{Q'} \leftarrow \theta^Q, \theta^{\mu'} \leftarrow \theta^\mu$ and also initialize replay buffer $R$. **for** *t=1, ... ,T* **do**

    Clean the replay buffer **B**.

    /* $\mathbf{B} = (\mathbf{B_1}, ..., \mathbf{B_m}, ..., \mathbf{B_M})$; */

    /* $B^m$: on-policy data for the $m$th intersection */

    /* Generate on-policy data */

    $\mathbf{B} = GOD(t)$;

    **for** *episode=1, ..., 400* **do**

        **for** *m=1,..., M, Global* **do**

            **if** $m \neq Global$ **then**

                $LAU(\mathbf{B},m)$;// Update local agents

            **end**

            **if** *agent=Global* **then**

                $GAU(\mathbf{B})$;// Update the global agent

            **end**

        **end**

    **end**

**end**

---

---

**Algorithm 3** GOD (Generating On-policy Data)

---

/* Run one hour of simulation with noise $\eta$*/
Input:    $t$: timestamp
$\theta_m^\mu$: parameters for the $m$th actor network
$\theta_G^\mu$: parameters for the global actor network
Output: **B**: on-policy data
$\beta = 0.95^t$; rate for time decline
**for** $m=1, ... , M$ **do**
    Get $W_G^m$ from the global actor network with the parameters $\theta_G^\mu$;
    $W_G^m = \beta \times W_G^m$; $W_L^m$=1-$W_G^m$;
    **for** $l=1, ... ,3600$ **do**
        /* $\epsilon$: the probability of choosing to explore */
        /* $\eta_m$: noise for epsilon greedy exploration*/
        $p = $ random(0,1);
$$\eta_m = \begin{cases} 0, if\ p \le \epsilon, \\ random(-5,5), if\ p > \epsilon; \end{cases}$$
$$a_l^m = \begin{cases} \mu(s_l|\theta_m^\mu) + \eta_m, & \text{if } W_L^m > W_G^m, \\ \boldsymbol{\mu}_G(s_l|\theta_G^\mu)(m) + \eta_m, & \text{if } W_L^m < W_G^m; \end{cases}$$
        Execute $a_l^m$ and observe $r_l^m, s_{l+1}^m$;
        Store transition $(s_l^m, a_l^m, r_l^m, s_{l+1}^m)$ in $\mathbf{B_m}$;

$\mathbf{B} = (\mathbf{B_1}, ..., \mathbf{B_m}, ..., \mathbf{B_M})$;
Return(B);

---

**Algorithm 4** LAU (Local Agent Updating)

---

Input:
**B**: on-policy data; $m$: the $m$th agent
$\theta_m^Q$: set of parameters for the local critic network;
$\theta_m^\mu$: set of parameters for the local actor network;
$(\theta_m^{Q'}, \theta_m^{\mu'})$: sets of parameters for the target network;
Output:
$\theta_m^Q$: new parameters for the $m$th critic network;
$\theta_m^\mu$: new parameters for the $m$th actor network;
$(\theta_m^{Q'}, \theta_m^{\mu'})$: new parameters for the target network;
Sample a random minibatch of $N_b$ transitions $(\boldsymbol{S}_i, \boldsymbol{A}_i, \boldsymbol{R}_i, \boldsymbol{S}_{i+1})$ from **B**;
Set $y_i^m = \boldsymbol{R}_i(m) + \gamma Q'(\boldsymbol{S}_{i+1}|\mu'(\boldsymbol{S}_{i+1}|\theta_m^{\mu'})|\theta_m^{Q'})$;
Update the critic parameters $\theta_m^Q$ by minimizing the loss: $L_{critic}^m = \frac{1}{N_b}\sum_i(y_i^m - Q(\boldsymbol{S}_i, \boldsymbol{A}_i|\theta_m^Q))^2$;
Update the actor parameters $\theta_m^\mu$ by minimizing the loss: $L_{actor}^m = -\frac{1}{N_b}\sum_i Q(\boldsymbol{S}_i, \mu(\boldsymbol{S}_i|\theta_m^\mu)|\theta_m^Q)$;
Update the target network:
$\theta_m^{Q'} \leftarrow (1-\tau)\theta_m^Q + \tau\theta_m^{Q'}$;
$\theta_m^{\mu'} \leftarrow (1-\tau)\theta_m^\mu + \tau\theta_m^{\mu'}$;

---

**Algorithm 5** GAU(Global Agent Updating)

---

Sample a random minibatch of $N_b$ transitions $(\boldsymbol{S}_i, \boldsymbol{A}_i, \boldsymbol{R}_i, \boldsymbol{S}_{i+1})$ from **B**;
Calculate $r_i^G$ by Eq.(17);
Set $y_i^G = r_i^G + \gamma Q_G'(\boldsymbol{S}_{i+1}|\boldsymbol{\mu}_G'(\boldsymbol{S}_{i+1}|\theta_G^{\boldsymbol{\mu}'})|\theta_G^{Q'})$;
Update the critic parameter $\theta_G^Q$ by minimizing the loss: $L_{critic}^G = \frac{1}{N_b}\sum_i(y_i^G - Q_G(\boldsymbol{S}_i, \boldsymbol{A}_i|\theta_G^Q))^2$;
Update the actor parameter $\theta_G^{\boldsymbol{\mu}}$ by minimizing the loss: $L_{actor}^G = -\frac{1}{N_b}\sum_i Q_G(\boldsymbol{S}_i, \boldsymbol{\mu_G}(\boldsymbol{S}_i|\theta_G^{\boldsymbol{\mu}})|\theta_G^Q)$;
Update the target networks:
$\theta_G^{Q'} \leftarrow (1-\tau)\theta_G^Q + \tau\theta_G^{Q'}$;
$\theta_G^{\boldsymbol{\mu}'} \leftarrow (1-\tau)\theta_G^{\boldsymbol{\mu}} + \tau\theta_G^{\boldsymbol{\mu}'}$;

# C PICTURE

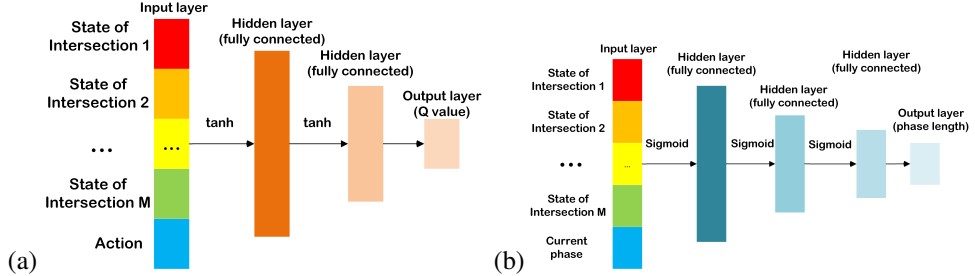

Figure 4: Architectures for local agent. (a) Local critic.(b) Local actor.

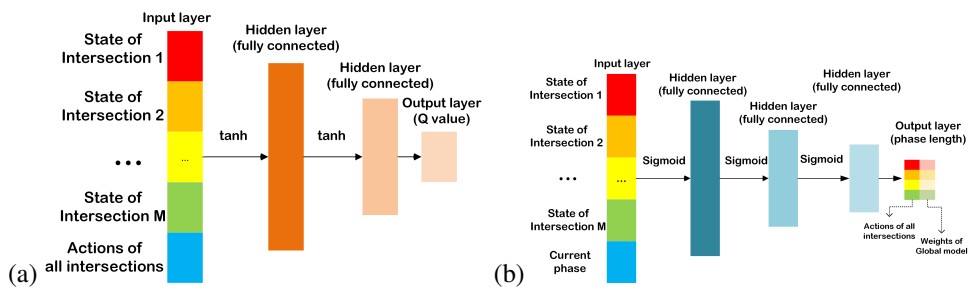

Figure 5: Architectures for global agent. (a) Global critic.(b) Global actor.

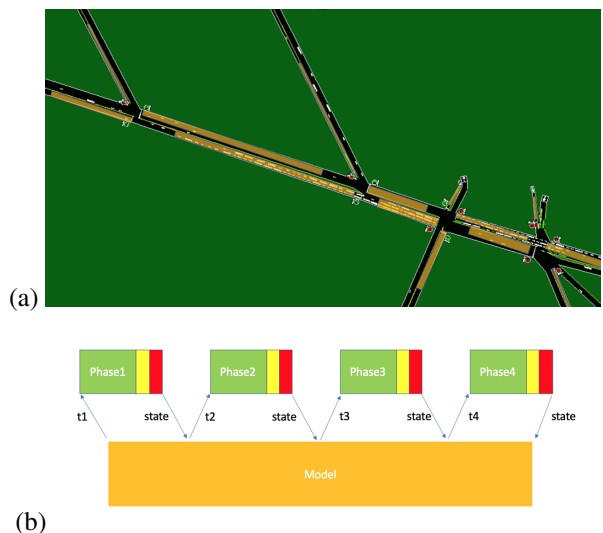

Figure 6: (a) Simulation traffic environment for RL traffic control, where yellow area is the visible range of each lane. (b) Control process. Here, $t_i$ means the duration of green light of phase $i$.

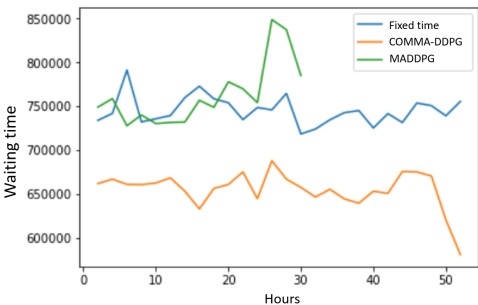

Figure 7: Waiting time converge conditions during the training process among different methods.

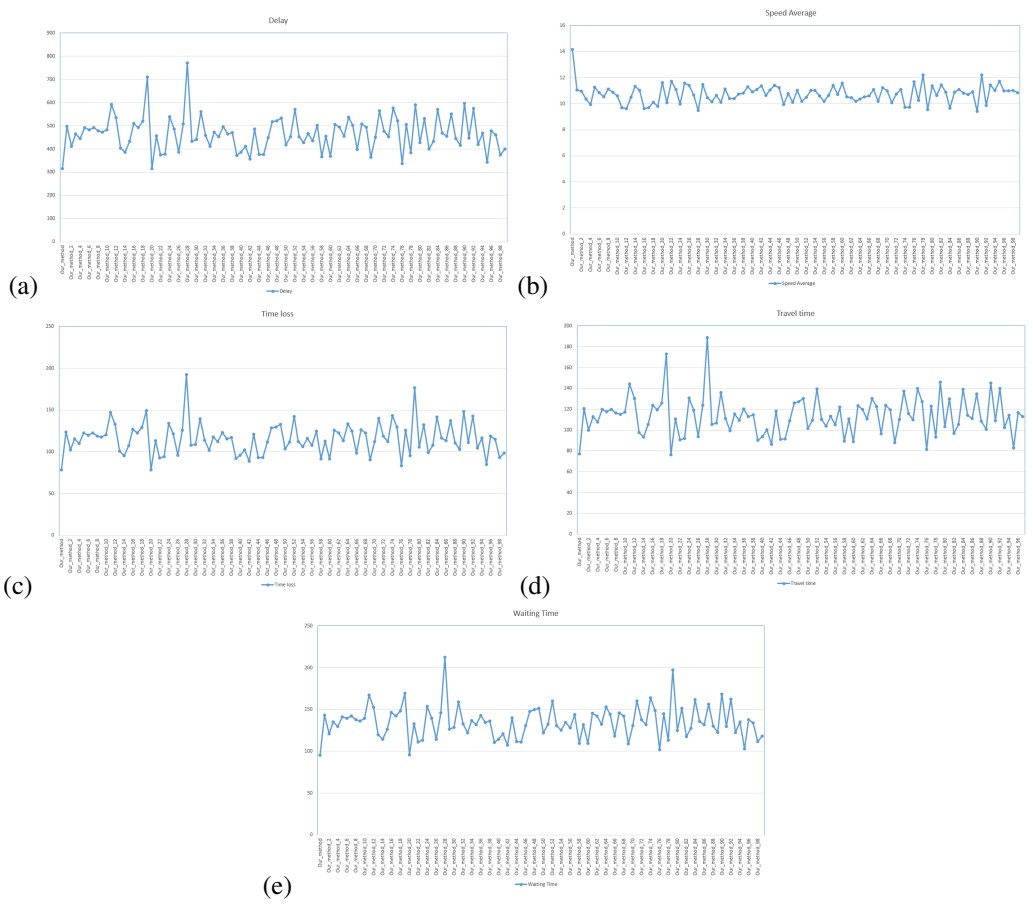

Figure 8: Figures (a) to (e) show the convergence of our method, where (a) is delay, (b) is speed, (c) is time loss, (d) is travel time, and finally (e) is waiting time

