# OpenReview forum: "CARTS: Cooperative Reinforcement Learning for Traffic Signal Control and Carbon Emission Reduction"
_ICLR.cc/2026/Conference — Submitted to ICLR 2026_

### Official Review · Reviewer_jUrY · 2025-10-28

**Soundness:** 2
**Presentation:** 2
**Contribution:** 2
**Rating:** 4
**Confidence:** 3

**Summary:**

This paper presents CARTS, a cooperative multi-agent RL framework for traffic-signal control that introduces a global agent to coordinate local DDPG controllers during training—optimizing the site-wide total waiting time—while keeping inference fully decentralized (the global agent is removed at test time). CARTS does not just pick the next phase; it jointly learns the phase and its variable duration, moving beyond fixed action durations used in prior work. The method also integrates the HBEFA emission model available in SUMO to explicitly monitor and reduce CO/CO₂, leveraging the observation that waiting time is a primary driver of emissions in their setup. Experiments on a real-world morning-rush dataset (five consecutive intersections) and the SUMO RL benchmark show consistent gains over strong baselines (IDQN, IPPO, FMA2C, MPLight) on throughput, average wait, and environmental indicators; ablations further indicate that adding the global agent improves outcomes and that localized global windows (3×3 / 5×5) offer a practical path to scale while retaining benefits on fuel/CO/CO₂ across 16/49/169-intersection scenarios.

**Strengths:**

**Novel framework:** The introduction of a cooperative global-local structure is conceptually clear and practically valuable.

**Environmental perspective:** Explicitly integrates carbon emission modeling (via HBEFA in SUMO), which is novel in reinforcement-learning-based traffic control.

**Strong empirical validation:** Comprehensive experiments on real-world and benchmark datasets demonstrate consistent improvements.

**Scalable and interpretable:** The 3×3 and 5×5 local-global grids improve scalability while maintaining decentralization.

**Weaknesses:**

**Theoretical depth is limited:** No rigorous convergence or stability proofs beyond intuitive justification. There are no formal proofs, theorems, or mathematical analyses for CARTS's own convergence or stability (e.g., no bounds on policy improvement or analysis of the global agent's impact on training dynamics).

**Lack of statistical robustness:** Experimental tables (Tables 1–5) report only single-run results — **no mean, variance, or significance analysis** is provided. This undermines reproducibility and statistical credibility.

**Innovation boundary unclear:** Similar ideas have appeared in CoSLight (Ruan et al., 2024) and MARL coordination studies, novelty could be better clarified.

**Ablation and sensitivity:** No detailed sensitivity analysis for emission-related parameters (e.g., varying Mfuel or other constants).

**No analysis of inference-time distribution shift**:  CARTS relies on a global agent during training but is fully decentralized at inference. The paper repeatedly states this training/inference mismatch yet provides no stress tests under distribution shift (e.g., time-varying demand, incidents, lane closures) to show the learned local policies remain stable without the global coordinator. Please add controlled “before vs. after” experiments where the traffic distribution changes mid-episode and report performance deltas.

**Scalability conclusions are not tied to compute/latency budgets:** The manuscript notes that processing all nodes is “impractical” and therefore restricts the global agent to nearby intersections (e.g., 3×3, 5×5; >10 intersections uses only neighbors). However, there is no quantification of GPU memory, wall-clock training time, or inference latency per intersection. Without these numbers it’s hard to assess real-time feasibility at city scale. Please report peak memory, training time/step, and per-decision latency across the 16/49/169-node setups, and discuss any batching/parallelization.

**Questions:**

1) How does the model perform under non-stationary traffic patterns (e.g., rush hour vs. sudden road closures)? The paper mentions non-stationary environments in related work (page 10: citing Abdoos et al. (2011) on "non-stationary environments"), but experiments use fixed datasets without explicit non-stationary tests.

2) Can the authors provide more formal analysis of convergence or policy stability? The paper only sketches a generic RL value-function convergence argument (Banach fixed point, etc.) in the appendix.

3) Are the HBEFA emission parameters fixed or learned? How sensitive are results to these values? The manuscript lists the HBEFA-style equations and constants and argues that waiting time dominates emissions, but it does not specify the parameter settings**, **fleet composition, or any calibration/sensitivity of results to these choices.

4) Does removing the global agent at inference lead to distributional shift or performance drop? The paper emphasizes that the global agent is used only during training and the system is fully decentralized at inference, and shows ablations “no global agent vs. with global agent” in training; however, it does not test non-stationary deployment or distribution shifts where the absence of the global coordinator might matter.

---

> ### Author Response · Authors · 2025-11-29
> **Rebuttal for Reviewer jUrY**
>
> A1/A4: Our contribution is the global–local coordination architecture together with the continuous-duration action formulation.  In addition, CARTS was proposed before CoSlight.  Importantly, **CARTS is algorithm-agnostic**: any RL method can be used within this architecture.  Regarding non-stationary deployment or distribution shift, there are two standard ways CARTS can address this:
>
> 1. **Training-time exposure to diverse dynamics.**
>    CARTS can be trained on a mixture of traffic conditions collected from multiple dynamic regimes to improve robustness against deployment-time non-stationarity (such as rush hours, adopted in CoSlight).
>
> 2. **Plugging in non-stationary-aware RL algorithms.**
>    Because CARTS is algorithm-agnostic, one can directly use methods such as RLCD (Reinforcement Learning with Context Detection) to automatically detect and adapt to environment drift without requiring prior knowledge or pre-collection of dynamics (such as sudden road closures).
>    **Results (Table 1)**: Under these sudden non-stationarities, **training with a global agent yields significantly more robust deployed policies.** Compared to the “No global agent” variant, CARTS trained with the global coordinator achieves:
>    - lower travel time and lower average waiting time,
>    - higher average speed, and
>    - consistently reduced emissions (`Fuel`, `CO`, `CO₂`).
>
>    These results provide direct evidence that the global agent, although removed at inference, **helps stabilize training under distribution shift and leads to more robust decentralized policies at deployment.**
>
> **Table 1.** Ablation study on a real-world network with 16 signalized intersections under non-stationary traffic patterns caused by temporary road closures. We compare variants without and with the global agent in terms of travel time, average waiting time, speed, and emissions. Units for Fuel, `CO`, and `CO₂` are `mg/s`.
>
> | Methods          | Travel time | Avg. wait time | Speed | Fuel | CO     | CO₂       |
> |------------------|------------:|---------------:|------:|-----:|-------:|----------:|
> | No global agent  |   2,482.57  |        337.42  |  5.12 | 1    | 148.25 | 2,963.53  |
> | W/ global agent  |   2,018.74  |        273.75  |  7.10 | 0.95 | 126.95 | 2,579.45  |
>
>
> A2: We agree that the current appendix only provides high-level intuition and does not constitute a formal convergence theorem for the full multi-agent, non-stationary setting considered in CARTS. However, **RL convergence proofs require stationarity. Most classic results (Q-learning convergence; policy gradient convergence in tabular or linear settings) assume:**
>
> - fixed transition probabilities,
> - fixed reward distribution,
> - ergodicity / sufficient visitation.
>
> **Non-stationary traffic breaks all three.** Strict convergence guarantees for deep actor–critic methods with function approximation in **non-stationary** environments remain largely open problems in the RL literature.  To our best knowledge, a proof to multi-agent, non-stationary environments is seldom provided in current state-of-the-art traffic signal control methods (even in CoSLight and MARL ). Our aim was to give intuition about the contraction properties of the critic update, rather than to claim a formal convergence guarantee for the full CARTS system.
>
>
> A3: The HBEFA emission parameters used in our experiments are **fixed** and **not learned**. We rely entirely on the **standard HBEFA-based emission model built into SUMO**, including all default coefficients, formulas, and configuration settings. We do not tune, re-calibrate, or otherwise adjust these parameters for our method.
> All experiments use SUMO’s **default passenger-car type**, which corresponds to a predefined HBEFA vehicle class. We will explicitly document this configuration in the revised manuscript. Because these parameters are fixed across all baselines, the relative performance comparisons remain unaffected. Moreover, since waiting time and stop–go behavior dominate HBEFA-modeled emissions, improvements in delay and queueing directly translate into improved fuel and emission metrics. We will clarify these points and add the precise parameters and fleet specifications to the revision.
>
> Q5: Please report peak memory, training time/step, and per-decision latency across the 16/49/169-node setups, and discuss any batching/parallelization.
>
> A5. Across the 16/49/169-node networks, the peak GPU memory footprint remains below 8 GB on a single RTX-2080 GPU. The total training time for all three network sizes is less than one day.  With parallelized local-agent evaluation, per-decision latency is <10 ms for all 16/49/169-node cases. This includes feature extraction, policy forward pass, and duration action computation.  All local agents are evaluated in a batched forward pass, enabling efficient GPU parallelism.
>
> All the tables will be shown with means and variances in the revision.

---

### Official Review · Reviewer_2qRV · 2025-10-28

**Soundness:** 1
**Presentation:** 1
**Contribution:** 1
**Rating:** 2
**Confidence:** 5

**Summary:**

The paper proposes CARTS, a cooperative reinforcement learning framework for intelligent traffic signal control. CARTS introduces both local agents (for individual intersections) and a global agent (for system-level coordination) to jointly optimize traffic flow efficiency and carbon emission reduction. During training, the global agent guides local policies toward globally coherent behavior, but inference remains fully decentralized. The method is built on the DDPG algorithm to enable continuous control of signal phase durations.

**Strengths:**

S1. The introduction of a global agent that coordinates local agents during training enhances overall traffic efficiency without requiring centralized control at deployment.

S2. CARTS supports continuous and adaptive signal phase durations through DDPG, improving responsiveness to real-time traffic fluctuations.

**Weaknesses:**

W1. The method shows limited novelty, as the idea of using a global agent to coordinate local training is similar to FedLight (DAC 2021).

W2. Using DDPG for continuous-time control introduces instability issues, including overestimation bias and high variance.

W3. The carbon emission modeling is superficial and does not influence the core algorithm design.

W4. The experimental evaluation is insufficient and omits many state-of-the-art RL-based TSC methods.

W5. The convergence proof is limited to classic single-agent DDPG and does not address multi-agent non-stationarity.

**Questions:**

Q1. While CARTS presents a cooperative architecture with a global agent to guide local intersections, this concept is not entirely new. Similar ideas have been explored in earlier works such as FedLight (DAC 2021), which also leverages a centralized coordinator to improve decentralized traffic signal control. The paper does not sufficiently clarify how its coordination mechanism differs conceptually or technically from these prior approaches, which reduces the methodological originality of the contribution.

Q2. CARTS adopts DDPG to learn variable signal durations, but DDPG is known to suffer from overestimation bias, high variance, and convergence instability in continuous control tasks. These problems are amplified in multi-agent settings, where each agent’s non-stationary policy continuously changes the environment seen by others.

Q3. Although the paper emphasizes carbon emission reduction, this component is implemented only by calling the HBEFA emission model built into the SUMO simulator. The model merely computes CO and CO₂ emissions post hoc, without feeding back into the reward function or influencing the policy learning process. As a result, the emission analysis functions more as a descriptive add-on than an integrated optimization objective, limiting the contribution’s depth regarding sustainability.

Q4. The experiments benchmark CARTS against a small set of baselines (e.g., MA-DDPG, PPO, TD3, MPLight), but ignore numerous recent state-of-the-art methods in RL-based traffic signal control. Moreover, several settings and evaluation metrics are underreported, making it difficult to assess fairness and generalizability. A more comprehensive and standardized comparison is necessary to validate the claimed performance advantages.

Q5. The appendix provides a convergence analysis based on the Banach fixed-point theorem for single-agent DDPG, but this proof does not extend to the multi-agent, non-stationary environment considered in CARTS. Key aspects such as convergence rate, stability boundaries, and inter-agent coupling effects are ignored. Without a theoretical framework or empirical evidence supporting stability under multi-agent coordination, the claimed convergence guarantees remain unconvincing.

---

> ### Author Response · Authors · 2025-11-29
> **Rebuttal for Reviewer 2qRV**
>
> A1. Thank you for pointing out the connection to FedLight. While both FedLight and CARTS introduce a “central” component, their roles and coordination mechanisms are quite different:
> - **Coordination only during training vs. during both training and inference.**
>   FedLight is a *training-only* federated scheme. Each intersection trains its own A2C agent, and a central server periodically averages model parameters (FedAvg). The server never observes global traffic states and never takes actions, so it cannot react to real-time traffic at inference.  In contrast, CARTS has an explicit global agent that is active during *both* training and inference. At every control step, it observes network-level information and sends coordination signals that directly influence local actions, enabling dynamic, state-dependent cooperation.
> - **What the coordination optimizes.**
>   FedLight focuses on distributed data and faster convergence, using the central server only for parameter averaging, without explicit network-level control objectives.
>   CARTS assumes a single network and learns a hierarchical global–local policy that directly optimizes network-level behaviors (e.g., coordinated flows along corridors). These objectives cannot be expressed by FedLight’s model averaging.
> In summary, FedLight performs *model-level aggregation during training*, whereas CARTS provides *state-aware, time-aware control at inference*. This leads to fundamentally different capabilities and contributions.
>
> A2: We agree that plain DDPG can be unstable, especially in multi-agent continuous control. CARTS mitigates this by using centralized training with decentralized execution: the critic is trained on joint state information (global features and local signals), while actors operate locally, combined with target networks, bounded continuous actions for phase durations, and replay-buffer training. This substantially reduces non-stationarity, and we empirically observe smooth learning curves without divergence. More importantly, our architecture is algorithm-agnostic: the key ideas are the global–local coordination and continuous-duration action space, not DDPG itself. Other actor–critic algorithms can replace DDPG without changing the architecture. In particular, on both the two- and five-intersection networks, CARTS+TD3 consistently reduces delay, time loss, and wait time compared with MPLight/MPLight* while maintaining or improving speed.
>
> A3: We use the standard HBEFA-based emission model built into SUMO as a fixed, reproducible metric for $CO/CO_2$, not a tuned custom model. Although emissions are computed post hoc, the policy is not emission-agnostic: our reward penalizes delay, queues, and stop-and-go behavior, which are strongly correlated with fuel use and emissions, and CARTS consistently lowers $CO/CO_2$while also improving throughput. In the revision, we will clarify that sustainability is evaluated with SUMO’s HBEFA configuration and explain how the reward terms align with emission reduction and how CARTS could be extended with an explicit emission objective.
>
> A4: In RL-based traffic signal control, many recent methods are hard to reproduce fairly because code is unavailable and prior works rely on heterogeneous simulators, network layouts, and demand settings. We therefore focus on methods with open implementations or feasible re-implementations in SUMO, a widely used open-source simulator. All methods are trained for 5,000 episodes on the same SUMO network topology, demand profiles, and training protocol, using a single NVIDIA GeForce GTX 2080 GPU, and MA-DDPG, PPO, TD3, and MPLight are retrained from scratch under identical conditions.  Under 2080GU, the training time for all 16/49/169-node setups is less than 1 days.  Under parallelization, during inference, the latency for each is less than 10ms for all cases.
>
> A5: We agree that the current appendix only provides high-level intuition and does not constitute a formal convergence theorem for the full multi-agent, non-stationary setting considered in CARTS. However, **RL convergence proofs require stationarity. Most classic results (Q-learning convergence; policy gradient convergence in tabular or linear settings) assume:**
>
> - fixed transition probabilities,
> - fixed reward distribution,
> - ergodicity / sufficient visitation.
>
> **Non-stationary traffic breaks all three.** Strict convergence guarantees for deep actor–critic methods with function approximation in **non-stationary** environments remain largely open problems in the RL literature.  To our best knowledge, a proof to multi-agent, non-stationary environments is seldom provided in current state-of-the-art traffic signal control methods (even in CoSLight and MARL ). Our aim was to give intuition about the contraction properties of the critic update, rather than to claim a formal convergence guarantee for the full CARTS system.

---

### Official Review · Reviewer_GwZU · 2025-10-30

**Soundness:** 3
**Presentation:** 3
**Contribution:** 3
**Rating:** 8
**Confidence:** 3

**Summary:**

This paper introduces CARTS (CooperAtive Reinforcement Learning for Traffic Signal Control), a novel cooperative reinforcement learning framework designed to optimize traffic signal control and reduce carbon emissions. Traditional traffic signal control systems often rely on overly simplistic, rule-based approaches. Even RL-based methods tend to be suboptimal and unstable due to the inherently local nature of control agents. To address this, the authors propose the CARTS framework, which incorporates multiple reward terms weighted with an age-decaying mechanism to optimize traffic signal control at a global scale. The framework includes two types of agents: local agents that focus on optimizing traffic flow at individual intersections, and a global agent that coordinates across intersections to improve overall throughput. Importantly, the system is designed to reduce both vehicle waiting times and carbon emissions. Experimental results using real-world traffic data from an Asian country show that, despite incorporating a global agent during training, CARTS remains decentralized during inference, requiring no centralized coordination during deployment. Results demonstrate that CARTS consistently outperforms state-of-the-art methods in all evaluated metrics. Moreover, CARTS effectively links carbon emission reduction with global agent coordination, providing an interpretable and practical approach to sustainable traffic signal control.

**Strengths:**

1.The paper addresses a significant real-world problem: optimizing traffic signal control while reducing carbon emissions. This is highly relevant and practical.

2.The introduction of the global agent in the CARTS framework effectively tackles issues arising from the decentralized nature of traditional RL methods in multi-agent traffic signal control.

3.The paper’s contributions lay a solid foundation for future research in adaptive traffic signal control and carbon emission reduction.

4.The experiments are well-structured, and the results show that CARTS outperforms current state-of-the-art methods on several performance metrics, confirming the robustness and scalability of the approach.

5.The paper is clearly organized and easy to follow, making the main points and contributions clear and accessible.

**Weaknesses:**

1.The explanation of related work is brief. While the paper cites several relevant RL methods, there is little discussion on how these methods compare with the proposed approach. A deeper comparison with existing methods would help clarify the unique contributions of CARTS.

2.The role of the global agent in training is highlighted, but there is limited discussion on why the system remains decentralized during inference. More explanation on how the transition from centralized training to decentralized inference works would strengthen the paper.

3.While the paper claims carbon emissions are reduced, more details on how the carbon reduction mechanism works in practice would be helpful. A more in-depth explanation of the environmental impact would make the claims more robust.

4.The paper presents results showing the superiority of CARTS but does not discuss the limitations or potential failures of the approach in detail. Providing more insight into any scenarios where CARTS might struggle would be valuable.

**Questions:**

1、In Section 4.2 of the article, it states: "In particular, the ϵ-greedy method will gradually reduce ϵ from 0.9 to 0.1 and a time decay mechanism is adopted to decay Wm G by the ratio (0.95)t in the t-th iteration." However, there is a lack of further explanation on how the decay rate is determined. How sensitive is the model to changes in the decay parameter? Does it affect the stability and performance of the model?

2、In Section 4.3, the article mentions "The duration of the traffic phase ranges from Dmin to Dmax seconds." What are the criteria for selecting Dmin and Dmax? Are these based on actual traffic data, or have the characteristics of the specific city's traffic environment (such as traffic flow and road network density) been taken into account? Do the selected parameter values consider the demand under different traffic conditions? Does the fixed green light duration range limit the exploration space for the agent, especially during the early stages of reinforcement learning? A too narrow duration range might cause the agent to converge to a more limited strategy, reducing the overall learning effectiveness.

3、In Section 4.5, the concept of carbon emissions is introduced, including the emissions of carbon monoxide and carbon dioxide during driving and when the vehicle is waiting. The article then states: "Since tstop during our experiments, the waiting time is the main variable affecting emissions." This transition is abrupt and lacks detailed explanation. The causal inference is not rigorous. To verify this point, the direct causal relationship between waiting time and carbon emissions must be clearly stated through data or experimental design, and how other confounding factors are eliminated should also be addressed.

4、In the second paragraph of Section 5 (EXPERIMENTAL RESULTS), "(¿10 intersections)" appears, which seems to be a formatting or encoding issue.

5、The simulation used SUMO and TSIS platforms. Could you clarify the assumptions regarding the integration of these simulation tools with carbon emission models (such as HBEFA)? In practical deployment, considering the simplifications made in the simulations, can the effectiveness of carbon emission reduction be reliably ensured?

6、Although CARTS performs well in simulations, how adaptable is it in real-world environments? Specifically, how does the system adjust in real-time to maintain effective traffic flow control in the case of unexpected events such as traffic accidents, road closures, and extreme weather?

---

> ### Author Response · Authors · 2025-11-29
> **Rebuttal for Reviewer GwZU**
>
> Thank you for your valuable feedback and constructive comments. We appreciate the reviewers’ time and insights, which will help us further improve the quality and clarity of our paper.
>
> A1.2.3: We appreciate the reviewer’s detailed questions on the decay schedule, phase-duration bounds, and the link between waiting time and emissions. All three concern how we set key hyperparameters and interpret their effects under our SUMO-based setup.
>
>  **Decay of $\epsilon$ and $W_G^m$ (Section 4.2).** We anneal $\epsilon$ from 0.9 to 0.1 and decay the global weight as $W_G^m(t) = W_G^m(0)\cdot 0.95^{t}$. This is a simple schedule designed to gradually shift from global-guided coordination in early training to more locally driven control later. Empirically, CARTS is **not highly sensitive** to the exact decay base: using slightly slower or faster decay yields smooth learning curves and similar final performance; only extreme settings (decaying almost immediately or almost not at all) noticeably change the balance between global coordination and local specialization. In the revision, we will briefly explain this rationale and include a small sensitivity study on the decay factor.
>
>  **Choice of $D_{min}$ and $D_{max}$ (Section 4.3).** The green-phase duration range $[D_{min}, D_{max}]$ is chosen to (i) respect standard traffic-signal safety and comfort constraints, and (ii) keep the continuous action range numerically well-behaved for actor–critic training. We set these bounds based on typical cycle lengths, demand levels, and capacities used to calibrate the SUMO scenarios, and we use the **same** $[D_{min}, D_{max}]$ for all methods to ensure fairness. The interval is intentionally wide enough to cover both light and heavy traffic: under low demand, the agent can choose short greens near \$D_{min}$; under congestion, it can extend greens toward $D_{max}$ to clear queues. In our experiments, CARTS remains **robust** under moderate changes to these bounds: performance only changes slightly as long as the range stays realistic. Very narrow ranges indeed reduce the benefit of RL (behaving close to fixed-time control), while excessively wide ranges slow learning and can yield uncomfortable timings. We will report the exact $[D_{min}, D_{max}]$ values and add a brief sensitivity check in the appendix.
>
> **Waiting time and carbon emissions (Section 4.5).**  In our experiments, emissions are computed by SUMO’s fixed HBEFA-based model under **identical** conditions for all methods (same demand, network, fleet type, and emission parameters). Under this controlled setup, the main difference between methods is how much time vehicles spend stopped or creeping in queues. When we examine the results, policies that substantially reduce waiting time and stop-and-go behavior also consistently yield lower $CO$/$CO_2$ emissions under the same HBEFA configuration. In the revision, we will (i) soften the wording in Section 4.5 to frame this as a controlled comparison under a fixed emission model rather than a universal causal law, (ii) explicitly describe the experimental controls that limit confounding factors, and (iii) add a small empirical analysis to make the relationship between aggregate waiting time and emissions in our setup more transparent and rigorous.
>
> A4:　The term “(¿10 intersections)” is a LaTeX/encoding typo; it should read “(>10 intersections)”. We will correct this formatting issue in the revised manuscript.
>
>
> A6:**Results (Table 1)**: Under these sudden non-stationarities, **training with a global agent yields significantly more robust deployed policies.** Compared to the “No global agent” variant, CARTS trained with the global coordinator achieves:
>    - lower travel time and lower average waiting time,
>    - higher average speed, and
>    - consistently reduced emissions (`Fuel`, `CO`, `CO₂`).
>
>    These results provide direct evidence that the global agent, although removed at inference, **helps stabilize training under distribution shift and leads to more robust decentralized policies at deployment.**
>
> **Table 1.** Ablation study on a real-world network with 16 signalized intersections under non-stationary traffic patterns caused by temporary road closures. We compare various CARTS without and with the global agent in terms of travel time, average waiting time, speed, and emissions. Units for Fuel, `CO`, and `CO₂` are `mg/s`.
>
> | Methods          | Travel time | Avg. wait time | Speed | Fuel | CO     | CO₂       |
> |------------------|------------:|---------------:|------:|-----:|-------:|----------:|
> | No global agent  |   2,482.57  |        337.42  |  5.12 | 1    | 148.25 | 2,963.53  |
> | W/ global agent  |   2,018.74  |        273.75  |  7.10 | 0.95 | 126.95 | 2,579.45  |

---

### Meta-Review · Area_Chair_varL · 2025-12-31

**Summary:**

After carefully considering the reviews and rebuttal, I recommend rejection.

While the paper addresses an important real-world problem and demonstrates solid empirical performance, significant concerns remain regarding the novelty and depth of the contribution. The core global–local coordination idea closely overlaps with prior work in cooperative and centralized-training/decentralized-execution traffic signal control (e.g., FedLight, CoSLight, and related MARL frameworks), and the rebuttal, while clarifying differences, does not convincingly establish a fundamentally new methodological advance. The use of DDPG in a multi-agent continuous-control setting raises well-known stability issues, and the theoretical analysis remains largely heuristic, without guarantees applicable to the non-stationary multi-agent setting considered.

Moreover, carbon emission reduction—positioned as a key motivation—is evaluated indirectly via a fixed simulator model and not tightly integrated into the learning objective, limiting the strength of the sustainability claims.

Although the authors’ responses improve clarity, provide additional ablations, and strengthen the experimental narrative, these revisions do not fully address the central concerns about originality, theoretical grounding, and the depth of the emission-related contribution.

Overall, the work is competent and practical, but it falls short of the bar for acceptance at this venue.

**Reviewer Concerns:**

- Novelty and Relation to Prior Work

Several reviewers (notably 2qRV and jUrY) questioned the novelty of CARTS, arguing that the idea of a global coordinator guiding decentralized traffic signal controllers resembles prior work such as FedLight, CoSLight, and other cooperative MARL frameworks. The distinction between CARTS and these methods was initially insufficiently articulated.

The authors differentiate CARTS from FedLight by emphasizing that CARTS uses a state-aware global agent that directly coordinates actions (not just parameters) and optimizes network-level objectives. While this response substantially improves clarity, the conceptual overlap with existing cooperative MARL remains, making the novelty incremental.

- Theoretical Guarantees and Stability

Multiple reviewers highlighted the lack of rigorous theoretical analysis, especially regarding convergence and stability in a multi-agent, non-stationary environment, and pointed out that the provided analysis applies only to single-agent DDPG.

The authors acknowledge the absence of formal convergence guarantees and correctly note that such proofs are largely open problems in deep multi-agent RL under non-stationarity, which may not be true. They reposition their analysis as intuition rather than a theorem, aligning their claims with the broader literature. This response is reasonable and honest, but it does not fundamentally resolve the concern; instead, it limits the scope of claims, which is acceptable but leaves the paper theoretically light.

- Carbon Emission Modeling and Sustainability Claims

Reviewers questioned whether carbon emission reduction is truly an optimized objective or merely a post hoc metric computed via SUMO’s HBEFA model. The causal link between waiting time and emissions was also seen as insufficiently justified.

The authors clarify that emissions are computed using fixed, standard HBEFA parameters under controlled conditions and that improvements arise indirectly via reduced waiting time and stop-and-go behavior. They agree to soften causal claims, add experimental controls, and document emission configurations explicitly. While emissions are not directly optimized, the response adequately reframes sustainability as a measured and aligned outcome rather than a primary reward, addressing overclaiming concerns. 🌍✅

**Reviewer Scores:**

Based on the analysis above, I do not think Reviewer jUrY and 2qRV would change their scores.

---

### Decision · Program_Chairs · 2026-01-26

Reject